# Adenosine-generating CD39⁺ plasmablasts predispose to successful infliximab therapy in pediatric IBD

Alexander Schnell[1], Benedikt Schwarz[1], Hannah Schmidt[2], Ida Allabauer[1], Wolfgang Schuh[3], Adrian P Regensburger[1], Manfred Rauh[2], Joachim Woelfle[2], André Hoerning[1]

**B cells display several immunoregulatory mechanisms including the production of interleukin-10. Ectonucleotidases like CD39 and CD73 influence immune homeostasis by metabolizing eATP and generating immunosuppressive adenosine. The major objective was to examine the expression of those immunoregulatory molecules on B-cell subsets, and, more specifically, to determine their association with an infliximab (IFX) treatment in a pediatric inflammatory bowel disease (IBD) cohort. 42 IBD patients were assessed for IFX response after 12 mo of therapy and compared against 14 healthy controls (HC). Although IL10-producing plasmablasts were decreased in IFX nonresponders (NRS), we detected an up-regulation of CD39 on plasmablasts and increased fractions of CD39/CD73-co-expressing naïve and memory B cells in responding patients (RS). In addition, B cells of responders proved to have superior ATP degradation capacities and adenosine production before therapy initiation compared with NRS and HC. Moreover, IFX nonresponders had a marked deficiency of α4β7hi plasmablasts, whereas both cohorts had fewer CCR9-expressing plasmablasts. Consequently, CD39⁺ plasmablasts were decreased in biopsies of inflamed mucosal tissues, especially in IFX nonresponders. Our results highlight the regulatory potential of CD39/CD73-expressing B cells in pediatric IBD and suggest CD39⁺ plasmablasts as a potential determinant of a successful immunosuppressive therapy with IFX.**

## Introduction

Inflammatory bowel disease (IBD) is considered to appear in individuals with genetic susceptibility and dysfunction of epithelial and mucosal immune processes in the context of environmental factors like infectious, microbial, or dietary triggers (1). IBD usually is classified as either Crohn's disease (CD), ulcerative colitis (UC), or IBD-unclassified (IBD-U). In contrast to adults, definitive diagnosis of either CD or UC in the pediatric cohort is challenging as children often display overlapping features of both entities (2, 3). Therapeutically, the introduction of TNF-antagonizing agents like infliximab (IFX) or adalimumab has revolutionized the treatment and has become a first-line approach in the case of pediatric IBD for severe CD and UC alike (4). However, primary nonresponse or secondary loss of response with/without the development of neutralizing antidrug antibodies still complicates the long-term effectiveness of these regimens (5). Although the role of T cells in the pathogenesis of IBD and in response to a TNF-antagonistic treatment has been extensively researched (6, 7), there is also a growing body of evidence that describes an involvement of B cells. We and others have reported that the numerical distribution of B cells in blood (8) and mucosa (9, 10) is perturbed in IBD patients and is altered in the course of an IFX treatment (11, 12). Apart from a potential pathologic role of B cells in IBD and in regard to the IFX therapy success, there are certain immunosuppressive mechanisms by which B cells are thought to regulate the immunological milieu in inflammatory diseases (13, 14).

Previous studies demonstrate that IL10-producing B cells exhibit suppressive capacities and these subsets are enriched in the CD24hi CD38hi transitional B-cell compartment (15); however, several other B-cell subsets like CD24⁺ CD27hi and CD38hi plasmablasts are competent IL10 producers. Moreover, B cells have been shown to secrete granzyme B (GrzmB) (16), a proapoptotic serine protease. B cell–derived granzyme B effectively limits T-cell responses (17) and is also able to degrade components of the extracellular matrix, thus actively shaping the local milieu.

In addition to the IL10- and GrzmB-mediated immunoregulation, there are several lines of evidence emphasizing the importance of purine metabolites as a crucial suppressive factor playing an important role of regulatory cell–mediated alteration of immune cell activity (18, 19, 20, 21). One of the most potent metabolites is adenosine (ADO), which is hydrolyzed from ATP in two subsequent steps by the purinergic enzymatic activity of the ectonucleoside triphosphate diphosphohydrolase-1 (NTPDase1) CD39 and the ecto-5'-nucleotidase CD73 (22, 23). In particular, several studies have

[1]Pediatric Gastroenterology and Hepatology, Department of Pediatrics and Adolescent Medicine, University Hospital Erlangen, Friedrich-Alexander-University Erlangen-Nuremberg, Erlangen, Germany   [2]Department of Pediatrics and Adolescent Medicine, University Hospital Erlangen, Friedrich-Alexander-University Erlangen-Nuremberg, Erlangen, Germany   [3]Division of Molecular Immunology, Department of Internal Medicine III, Nikolaus-Fiebiger Center, University Hospital Erlangen, Friedrich-Alexander-University of Erlangen-Nürnberg, Erlangen, Germany

Correspondence: alexander.schnell@uk-erlangen.de

**Table 1.** Demographic and clinical data of the study cohort.

| | Responder, n = 21 | Nonresponder, n = 21 | Healthy (n = 14) | *P*-value |
|---|---|---|---|---|
| **Demographic data** | | | | |
| Age (yr) | 13.9 (±2.6; range 7–17) | 13.2 (±2.9; range 6–17) | 12.3 (±5.2; range 6–23) | 0.44 |
| Sex | ♀: 12; ♂: 9 | ♀: 3; ♂: 18 | ♀: 5; ♂: 9 | 0.009 |
| BMI | 19.7 (±6.1) | 19.4 (±5.6) | | 0.88 |
| Entity | CD: 14; UC: 7 | CD: 16; UC: 5 | | 0.73 |
| **Clinical parameters** | | | | |
| Leukocytes ($10^3$/μl) | 8.6 (±2.9) | 9.0 (±3.6) | | 0.64 |
| CRP (mg/dl) | 20.3 (±23.6) | 13.8 (±28.2) | | 0.44 |
| ESR (mm/h) | 38.4 (±20.3) | 26.5 (±15.3) | | 0.09 |
| Hb (g/dl) | 11.5 (±1.8) | 11.7 (±1.5) | | 0.75 |
| IgG (g/l) | 12.8 (±4.3) | 10.7 (±2.5) | | 0.11 |
| Calprotectin (μg/g) | 11,354 (±21,479) | 2,369 (±2,570) | | 0.11 |
| Serum TNFα (ng/ml) | 33.1 (±64.7) | 10.2 (±2.7) | | 0.19 |
| **Medication** | | | | |
| Prednisolone (<0.1 mg/kg body weight) | 4 (1) | 9 (1) | | 0.11 |
| Azathioprine | 7 (8) | 9 (9) | | 0.75 |

Numbers in brackets indicate medication after 12 mo of IFX therapy.
Data are indicated as means where appropriate.

shown that ADO acts immunoregulatorily and inhibits the functions of different target cell populations and subsets of the immune system, including T and B lymphocytes, NK cells, dendritic cells, monocytes, and macrophages (19, 21, 24, 25, 26).

Importantly, recent data also suggest that especially B cells use purinergic signaling to control the activity of effector T cells and self-regulate their own function in an autocrine manner (20, 27).

Hence, the major objective of this study was to examine the expression of immunoregulatory molecules like IL10 and the ectonucleotidases CD73 and CD39 on B-cell subsets of pediatric IBD patients, and, more specifically, to determine their functional association with a successful or an unsuccessful IFX treatment in the longitudinal course in a pediatric IBD cohort.

# Results

### Demographic and clinical characteristics of the study population

We recruited 21 patients with clinical and biochemical responses (RS), whereas 21 patients were classified as NRS with respect to the endpoint criteria. A total of 34 of the 42 patients reached the study endpoint of 12 mo. Reasons for an earlier dropout were discontinuation of IFX treatment because of adverse drug effects in two patients (anaphylactic reaction after infusion, de novo IFX-induced psoriasis refractory to topic therapy) and proctocolectomy in one UC patient with a failing IFX therapy. In three additional patients, IFX treatment was switched to adalimumab because of antidrug antibody–mediated treatment failure. Two other patients left the study after 9 mo for nonmedical reasons, in one case categorized as NRS because of elevated fecal calprotectin and complete remission

in the other, therefore categorized as RS. The respective clinical data for each patient cohort are summarized in Table 1. No differences regarding the baseline characteristics were noted except for sex distribution.

### IFX responders display a decrease in circulating plasmablasts during IFX therapy

In order to identify differences in the composition of peripheral B-cell subsets, we analyzed the cohort with respect to the response status according to the gating strategy shown as described in previous studies (28, 29, 30, 31) (Fig 1A). Before initiation of the IFX therapy, we found a highly significant difference for naïve B cells, whose frequencies were elevated in patients with IBD and represented the predominant B-cell subset in peripheral blood in both RS and NRS alike (Fig 1B; RS: 71.4% [*P* = 0.0008], NRS: 71.7% [*P* = 0.0002], HC: 61.7%). For any other subsets, we could not determine a significant difference, although for circulating plasmablasts of IBD patients, we observed a trend toward higher frequencies, especially for RS in comparison with HC (Fig 1C; RS: 0.85% [*P* = 0.1], NRS: 0.56% [*P* = 0.61], HC: 0.35%). Interestingly, after 6 and 12 mo of therapy, the percentages of plasmablasts in RS declined significantly, reaching the percentages of plasmablasts in HC (Fig 1D, t6: *P* = 0.03, t12: *P* = 0.05).

### B cells of future nonresponders have a pro-inflammatory phenotype

As we detected only slight differences regarding B-cell frequencies between RS and NRS, we wanted to determine the (anti-)inflammatory cytokine production with respect to the response status of

**Figure 1. Gating strategy and numerical distribution of B cell subsets.**
**(A)** Gating strategy for unstimulated, CD19⁺ B cells. **(B)** Frequencies of peripheral B-cell subsets for responders (RS) or nonresponders (NRS) in comparison with HC before initiation of IFX. RS: left bars, vertically hatched. NRS: middle bars, diagonally hatched. HC: right bars, blank. T1: CD24^hiCD38^hi transitional cells; T2: CD24⁺CD38⁺

an IFX therapy. Because of low absolute B-cell numbers caused by small sample sizes in younger children and reduced B-cell numbers in the peripheral blood, we were not able to assess cytokine production for all patients at all time points as samples were split for analysis of B-cell subpopulations (Fig 1) and intracellular cytokine flow cytometry. In total, we analyzed samples of 9 patients throughout the complete study course (t0: n = 13 [RS: 6, NRS: 9], t3: n = 16 [RS: 10, NRS: 6], t6: n = 20 [RS: 11, NRS: 9], t12: n = 16, [RS: 10, NRS: 6]). We identified three subpopulations as previously described (11, 32): CD24$^{hi}$ CD38$^{hi}$ transitional B cells, CD38$^+$ mature B cells, and CD38$^{hi}$ plasmablasts. The gating scheme and representative flow cytometric analyses for stimulated B cells are shown in Fig S1.

Before initiation of IFX, we observed that NRS displayed reduced fractions of IL10$^+$ B cells within all analyzed subsets (Fig 2A and B, CD24$^{hi}$ CD38$^{hi}$: 21.1 ± 13% [NRS] versus 40.8 ± 11.4% [HC] [$P$ = 0.004], CD38$^{hi}$: 4.0 ± 3.4% [NRS] versus 19.8 ± 9.3 [HC] [$P$ = 0.03], CD38$^+$: 11.5 ± 7.8% [NRS] versus 25.8% [HC] [$P$ = 0.05]). We also detected a trend toward higher percentages of TNF$\alpha^+$ B cells within all subsets (Fig 2C, $P$ = 0.04), especially for NRS. Moreover, we analyzed the general polarization of B cells as measured by the logarithmized ratio of IL10 (anti-inflammatory)- to TNF$\alpha$ (pro-inflammatory)-producing B cells. As expected, in HC, CD24$^{hi}$ CD38$^{hi}$ transitional B cells displayed a pronounced IL10-skewed phenotype, whereas the CD38$^+$ core population consisting of mature B cells was rather balanced between IL10$^+$ and TNF$\alpha^+$ B cells (Fig 2D). In RS, all described B-cell subsets were balanced as well, yet with a larger individual variance than observed in HC (Fig 2D). However, the most striking differences were detected for B cells of NRS: whereas CD24$^{hi}$ CD38$^{hi}$ transitional B cells displayed a rather balanced ratio between IL10$^+$ and TNF$\alpha^+$ B cells (Fig 2D, HC: 0.7 ± 0.28 versus NRS: −0.01 ± 0.3, $P$ = 0.008), the CD38$^{hi}$ and the CD38$^+$ subsets were highly skewed toward TNF$\alpha$ production (Fig 2D, CD38$^{hi}$: 0.46 ± 0.25 [HC], 0.2 ± 0.49 [RS], −0.53 ± 0.49) (NRS [RS: $P$ = 0.003, HC: $P$ = 0.0002]; CD38$^+$: 0.04 ± 0.19 [HC] versus −0.53 ± 0.3) (NRS, $P$ = 0.03). In the course of IFX treatment, CD38$^{hi}$ plasmablasts of NRS displayed a sustained restoration of the initially perturbed cytokine polarization (Fig 2E).

To summarize, B cells of NRS display a rather pro-inflammatory phenotype before IFX initiation characterized by a significantly reduced ratio between IL10$^+$ and TNF$\alpha^+$ B cells. In the course of IFX treatment, the ratio of IL10-expressing to TNF$\alpha$-expressing cells within CD38hi plasmablasts normalizes for NRS.

### CD39 and CD73 are differentially expressed in peripheral blood B-cell subsets

We also wondered whether peripheral blood B cells expressed the ATP-degrading ectoenzymes CD73 and CD39 that have been shown to exert important immunoregulatory functions by degradation of ATP to adenosine (Ado) (33).

In general, we observed that the surface abundance of CD39 was constantly up-regulated along the transition of immature, transitional B cells toward naïve and antigen-experienced B cells like

memory B cells and plasmablasts in the blood, reaching its highest levels in plasmablasts (Fig 3A). In contrast to that, CD73 was only mainly expressed in naïve and memory B cells. A substantial amount of CD73 could still be detected in transitional B cells, yet at significant lower levels than in naïve B cells, whereas CD73 was absent in plasmablasts in the blood (Fig 3B). Consequently, a significant co-expression of both ectonucleotidases was only present in naïve and memory B cells as can be observed on the representative FACS plots shown in Fig 3C.

### CD39$^+$ CD73$^+$ B-cell subsets are increased in the blood of IFX responders at baseline and in the therapy course

To determine whether the surface abundance of CD39 and CD73 on B-cell subsets may be associated with the response status of an IFX therapy, we analyzed their expression before and during treatment in the longitudinal course. Before initiation of an IFX therapy, the surface abundance as assessed by the mean fluorescence intensity of CD39 was 1.3-fold higher in plasmablasts in RS compared with NRS (Fig 3A, $P$ = 0.01). On the other hand, the mean fluorescence intensity of CD73 was 3.5-fold higher in T2 transitional B cells in NRS compared with HC subjects (Fig 3B, $P$ = 0.002). As both CD39 and CD73 represent a functional axis and work together in extracellular ATP (eATP) degradation, we also analyzed the co-expression of both markers for each B-cell subset.

Here, we observed an elevation in the proportion of CD39$^+$/CD73$^+$ naïve and memory B cells in RS compared with HC (Fig 3C and D, naïve: 73.3 ± 9.4% [RS] versus 54.7 ± 22.9 [HC] [$P$ = 0.004]; memory: 49.3 ± 11.9 [RS] versus 31.1 ± 16.4 [HC] [$P$ = 0.005]) at baseline. Although the frequencies of CD39$^+$ CD73$^+$ B-cell subsets also tended to be higher in NRS compared with HC, it failed to reach the level of statistical significance (naïve: 65.0 ± 23.5 [NRS, $P$ = 0.16], memory: 41.9 ± 21.2 [NRS, $P$ = 0.15]).

In the longitudinal course of the IFX therapy, CD39$^+$- and CD73$^+$-co-expressing memory B-cell (Fig 3E) subpopulations of RS remained constantly elevated compared with HC subjects.

In summary, we detected an up-regulation of CD39 in responding plasmablasts and increased frequencies of CD39$^+$ CD73$^+$−co-expressing naïve and memory B cells in responding patients at the initiation of IFX therapy and during the course of the therapy.

### IFX-responding patients display a higher ATP-degrading activity

Because a significant amount of naïve and memory B cells of RS co-expressed the ectonucleotidases CD39 and CD73, we wondered whether the ATP-degrading functional axis of CD73 and CD39 on B cells was altered with regard to the IFX response status. To that account, CD19$^+$ B cells from patients already undergoing an IFX regimen for nine to 11 mo and HC (RS: n = 5, NRS: n = 6, HC: n = 4) were incubated in the presence of eATP for 1 h. Here, we detected that the normalized levels of eATP in the supernatant decreased to a higher extent in the assay for RS than for NRS or HC when B cells were

---

transitional cells; naïve: CD27$^-$ CD38$^+$ B cells; mem: CD27$^+$ CD38$^+$ B cells; pb: CD19$^+$ CD20$^-$ CD38$^{hi}$ plasmablasts. **(C)** Percentages of plasmablasts of RS (green) and NRS (red) compared with HC (gray). **(D)** Development of plasmablasts (D) of RS (green circles) and NRS (red squares) compared with HC (red dotted line). Level of significance is indicated by asterisks: *$P$ < 0.05; ** $P$ < 0.01; ***$P$ < 0.001; ****$P$ < 0.0001.

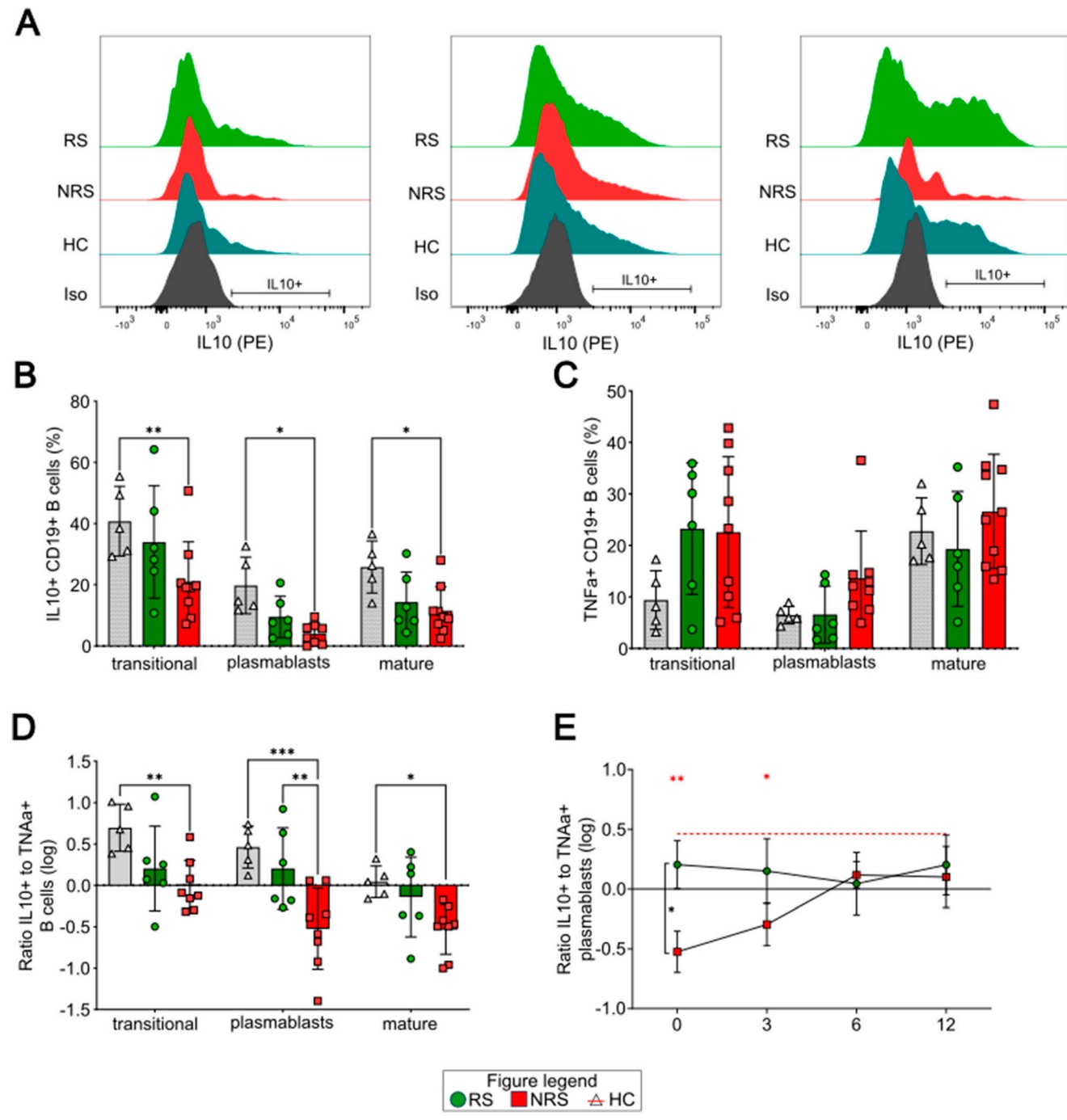

**Figure 2. Cytokine production in peripheral B cell subsets.**
**(A)** Representative histograms of IL10 stainings of transitional (left), plasmablasts (middle), and mature B cells (right panel) for RS (green), NRS (red), and healthy controls (turquoise). Isotype is shown in gray. **(B, C)** Fractions of IL10$^+$ (B) and TNFα$^+$ (C) B cells of future responders (RS; middle bar, green circles) or nonresponders (NRS; right bar, red squares) in comparison with healthy volunteers (HC; left bar, gray triangles) before initiation of IFX. **(D)** Ratio of IL10$^+$ to TNFα$^+$ B cells before IFX. **(E)** Course of the IL10$^+$/TNFα$^+$ ratio in CD38$^{hi}$ plasmablasts of RS (green circles) and NRS (red squares) compared with HC (red dotted line) in the course of IFX therapy. Level of significance is indicated by asterisks: *$P < 0.05$; **$P < 0.01$; ***$P < 0.001$; ****$P < 0.0001$.

stimulated (Fig 4A). Analysis of the corresponding area under the curve of the individual measurements revealed a significant difference between RS and NRS (Fig 4B, $P = 0.03$).

To validate those findings, we further analyzed the concentrations of the ATP products ADO and inosine (INO). In a first step, we

investigated the exact kinetics of this reaction in a preliminary experiment with CD19$^+$ B cells from HC. We observed a constant increase in the ADO concentration within the first 150 min, and there were no differences detectable between an unstimulated and stimulated condition. After 150 min, the ADO concentration

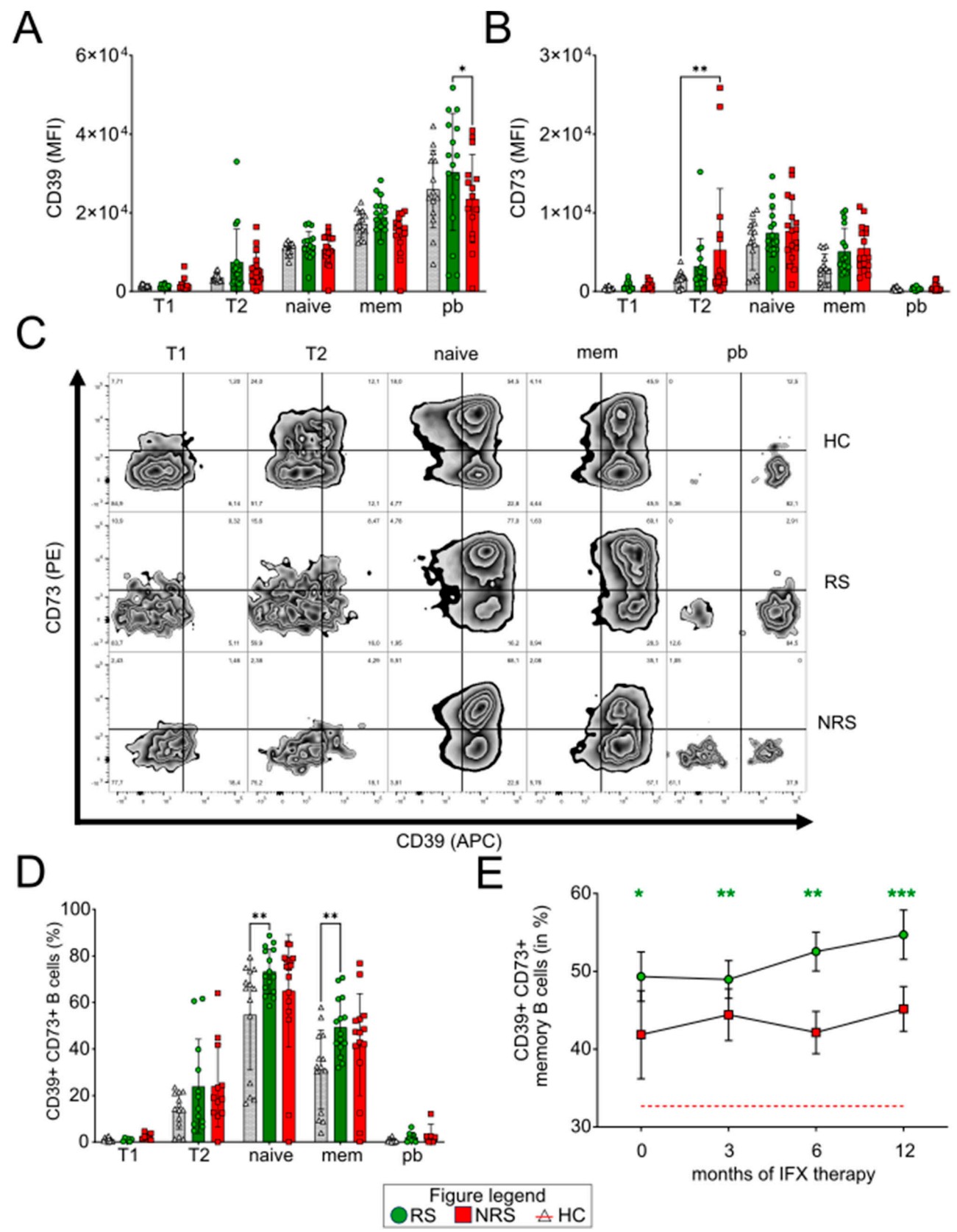

increased further for stimulated B cells, whereas it reached a plateau for unstimulated B cells (Fig 4C). Therefore, we focused on the time points 150 and 240 min for the further analysis of ADO and INO in the analysis of the patient samples.

We observed that even in unstimulated conditions, B cells of RS reached significantly higher concentrations of ADO than HC and tendentially also NRS (Fig 4D, 150 min: $0.65 \pm 0.55\ \mu mol/l$ [RS] versus $0.07 \pm 0.09\ \mu mol/l$ [HC, $P = 0.07$]; 240 min: $0.84 \pm 0.62\ \mu mol$ [RS] versus $0.12 \pm 0.08$ [HC, $P = 0.02$] versus $0.32 \pm 0.14\ \mu mol/l$ [NRS]). INO was only detectable at low concentrations in unstimulated conditions. Although the concentration was at least in trend higher than for NRS and HC, it failed to reach the threshold of statistical significance (Fig 4E). When B cells were stimulated, the concentration of ADO in general was higher. Also in this setting, RS tended to reach higher concentrations of ADO (Fig 4F, $1.5 \pm 1.2\ \mu mol/l$ [RS] versus $0.41 \pm 0.2\ \mu mol/l$ [HC, $P = 0.08$]) and INO (Fig 4G, $0.3 \pm 2.6\ \mu mol/l$ [RS] versus $0.24 \pm 0.02\ \mu mol/l$ [HC, $P = 0.03$]) than HC after 240 min.

Taken together, these data suggest that B cells of IFX-responsive children with IBD display a higher enzymatic activity of the degradation of ATP to ADO. In contrast, B cells of NRS patients only showed slightly increased ATP degradation capacity.

### Fractions of integrin α4β7hi and CCR9+ plasmablasts are decreased in IFX-unresponsive children

Gut homing of B cells with regulatory function is crucial especially if immunoregulation is mediated by cytokine secretion and/or local ATP-degrading properties to modulate the inflammatory mucosal microenvironment. Therefore, we wondered whether the fraction of mucosal homing receptor–expressing B cells was altered with respect to the IFX response status. To that account, the B-cell subsets described in Fig 1 were assessed regarding the expression of integrin (Itg) $\alpha 4\beta 7^{hi}$ and CCR9. The fractions of Itg $\alpha 4\beta 7$–expressing B-cell subsets were not significantly altered between IBD patients and HC and ranged from ~40% for memory B cells to ~80% for T2 transitional cells (data not shown). However, we noted that memory B cells and plasmablasts displayed a subset of Itg $\alpha 4\beta 7^{hi}$–expressing cells that was not apparent for naïve B cells (Fig 5A). The specific isotypes for each marker are shown in Fig 5B.

For these specific cell subsets, we observed significant differences in IBD patients: although at therapy initiation the fraction of Itg $\alpha 4\beta 7^{hi}$ memory B cells was significantly elevated for RS compared with HC (Fig 5C, $11.3 \pm 8.5\%$ [RS] versus $6.1 \pm 2.6\%$ [HC], $P = 0.006$), the percentage of Itg$\alpha 4\beta 7^{hi}$ plasmablasts was markedly reduced for NRS compared with HC and RS (Fig 5C, $22.5 \pm 9.5\%$ [NRS] versus $32.9 \pm 15.1$ [HC, $P = 0.003$], versus $38.3 \pm 9.0\%$ [RS, $P = <0.0001$]). During IFX therapy, the proportion of Itg $\alpha 4\beta 7^{hi}$ plasmablasts of NRS tended to increase slightly (data not shown), whereas the fraction of Itg $\alpha 4\beta 7^{hi}$ memory B cells of NRS increased significantly within the first 3 mo of IFX therapy (Fig 5D, t0: $8.3 \pm 4.0\%$ to t3: $11.2 \pm 6.3\%$, $P = $

0.04) and consolidated on the same elevated level as RS compared with HC.

For CCR9, a representative histogram is also shown in Fig 6A. We noted that the fraction of CCR9+ plasmablasts of IBD patients was significantly lower compared with HC (Fig 6B, $26.8 \pm 7.2\%$ [HC] versus $10.2 \pm 6.9\%$ [RS, <0.0001] versus $5.5 \pm 1.4\%$ [NRS, <0.0001]). Moreover, there was also a statistically significant difference between RS and NRS ($P = 0.04$). In the course of IFX therapy, the fraction of CCR9+ plasmablasts remained at a markedly reduced level throughout the whole therapy (Fig 6C).

In summary, we show that future IFX nonresponders display significantly decreased fractions of Itg $\alpha 4\beta 7^{hi}$ and CCR9+ plasmablasts compared with HC.

### Numbers of CD39-expressing mucosal plasmablast cells are decreased in inflamed mucosa of pediatric IBD patients

We also examined the appearance of CD39- or CD73-expressing CD19+ B cells in the mucosa of pediatric IBD patients (RS: n = 9 [CD: n = 4; UC: n = 5]; NRS: n = 16 [CD: n = 12; UC: n = 4]) before the initiation of the IFX therapy and compared these results with noninflamed tissues from HC subjects (n = 9) by immunohistochemical staining.

Unexpectedly, CD73+ B cells were scarcely present in the immunohistochemical stainings (Fig 7A, left panel). As only CD19+ CD39+ cells were detectable in the mucosa of IBD patients and HC (Fig 7A, right panel, encircled cells), we concluded that those cells were most likely plasmablasts. After counting, the numbers of CD39+ mucosal plasmablasts were markedly decreased in mucosal biopsies showing signs of active inflammation or chronicity (presence of neutrophils and/or signs of chronicity as defined by the presence of lymphoplasmacellular infiltrations), in the routinely performed pathological workups (Fig 7B).

Therefore, we focused on patient specimens that displayed features of high activity and found that CD39+ B cells were significantly reduced, but irrespective of the IFX response status (Fig S2A). In contrast to that, CD39− B cells were present in high abundance in the mucosa of IBD patients, especially in the ileum (Fig S2B). In order to better grasp the distinct composition of the mucosal B cells with regard to CD39 expression, we calculated the logarithmized ratio between mucosal CD39− and CD39+ B cells. In that context, a positive value represents a predominance of CD39+ B cells, whereas negative numbers mark a relative absence of CD39+ B cells. We observed that whereas HC displayed mostly CD39+ mucosal B cells irrespective of the localization, the ratio was significantly decreased in ileal biopsies taken from IBD patients and rather balanced within the colon and sigma, however still markedly reduced compared with HC (Fig 7C).

In summary, IBD patients with active mucosal inflammation displayed a relative deficiency of CD39-expressing plasmablasts,

**Figure 3. Subset-specific expression of the ectonucleotidases CD39 and CD73 in B cells.**
**(A, B)** Mean fluorescence intensities of CD39 (A) and CD73 (B) surface abundance for B-cell subsets of future responders (RS; left bars, vertically hatched, green) or nonresponders (NRS; middle bars, diagonally hatched, red) in comparison with healthy volunteers (HC; right bars, blank) before initiation of IFX. **(C)** Representative zebra plots of the CD39/CD73 co-expression in each B-cell subset and cohort. **(D)** Frequencies of CD39- and CD73-co-expressing B-cell subsets of future responders before initiation of IFX. **(E)** Co-expression of CD39+ CD73+ memory B cells during IFX therapy. T1: CD24hiCD38hi transitional cells; T2: CD24+CD38+ transitional cells; naïve: CD27− CD38+ B cells; mem: CD27+ CD38+ B cells; pb: CD19+ CD20− CD38hi plasmablasts. Level of significance is indicated by asterisks: *$P < 0.05$; **$P < 0.01$; ***$P < 0.001$; ****$P < 0.0001$.

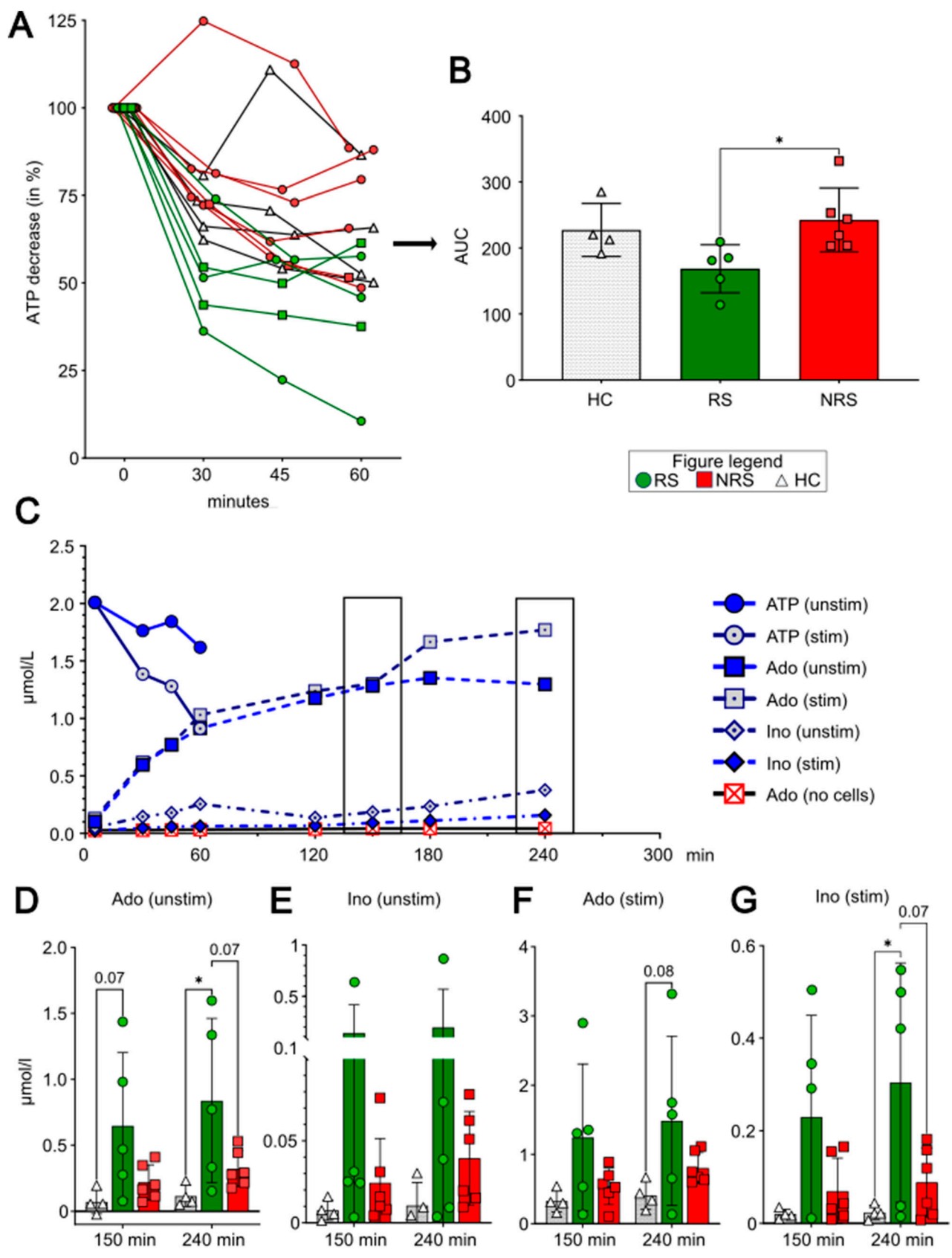

with the most prominent differences to the HC cohort in the terminal ileum.

# Discussion

The aim of this study was to elucidate the involvement of B-cell subsets in the context of an immunosuppressive therapy with IFX in pediatric IBD patients with particular interest in the functional analysis of the ATP-degrading ectoenzymes CD39 and CD73.

Although the composition of B cells of patients with IBD proved to be highly perturbed before initiation of an IFX treatment ([11]), we did not detect specific differences regarding the comparison of IFX-responding patients and nonresponsive patients. However, as formation of anti-IFX antibodies represents a major cause of an unsuccessful treatment leading to a secondary loss of response ([5]), our particular interest lay in peripheral antibody-secreting cells such as plasmablasts. In our analysis, we observed—at least in trend—higher percentages of plasmablasts in RS before IFX initiation compared with HC, which displayed a sustained normalization in the course of IFX treatment. Defendenti et al. also reported no relevant alterations of plasmablast frequencies in anti-TNF–treated patients, but the gating scheme used in their study differs strongly from ours ([34]).

Next, we phenotyped the major B-cell populations with regard to the production of TNFα and IL10, in order to characterize the overall inflammatory polarization of the respective subsets. Here, IFX nonresponsive patients displayed a profoundly pro-inflammatory skewed phenotype with reduced fractions of IL10+ B cells within all subsets, but most pronounced in the CD38hi plasmablast subpopulation. This observation could mainly be attributed to a relative deficiency of IL10+ B cells. Among all B-cell subsets, transitional B cells are thought to be the main producers of B cell–derived IL10 ([14], [15]), but also plasmablasts have been shown to produce relevant amounts of IL10 ([35]). In the context of IBD, deficiency or exhaustion of IL10+ regulatory B cells has been well established in the past years for CD ([36]) and UC ([37]) alike, highlighting the importance of IL10+ B cells in maintaining immunological homeostasis. However, our data suggest that also IL10-producing plasmablasts are involved in pediatric patients with IBD that prove nonresponsive toward IFX after 1 yr.

Furthermore, we observed an increase in the surface expression of CD39 plasmablasts and increased percentages of CD39/CD73-co-expressing naïve and memory B cells in responsive patients, as well as an increased B cell–mediated ATP degradation capacity and Ado generation in those patients. Purinergic signaling and deploying of ATP metabolites by unrestricted CD39 and CD73 activity appear to play a profound role in IBD with regard to experimental evidence as CD39-deficient ([38], [39], [40]) and CD73-deficient KO mice ([41], [42], [43])

show exaggerated features of chemically induced colitis. In humans, high CD39 expression in circulating Tregs has been shown to correlate with clinical remission in IBD patients, whereas single nucleotide polymorphisms, associated with low CD39 mRNA levels, increase the predisposition to Crohn's disease ([38]). To date, there are no reports on ectonucleotidase-expressing B cells or plasmablasts in current literature. However, Nascimento et al. described an expanded subset of CD39hi plasmablasts and increased adenosine concentrations in patients suffering from bacterial sepsis ([44]). This mechanism occurs most likely to dampen the detrimental consequences of systemic hyperinflammation during sepsis. However, we believe that in consideration of the previous data with regard to CD39 in IBD, this model can also be transferred to IBD, as patients with severe IBD might also display features of systemic inflammation (e.g., elevated ESR or CrP levels). In that context, our findings on the expression patterns of CD39 on plasmablasts and in addition CD73 on naïve and memory B cells of IBD patients together with the higher ATP-degrading capacity of B cells from IFX-responsive patients add valuable information on the importance of B cell–dependent ATP degradation or B cell–derived adenosine generation in IBD.

With respect to the IFX response, our data revealed a profound difference in Itg α4β7hi plasmablasts between RS and NRS. Indeed, Uzzan et al recently reported significantly increased proportions of β7+ plasmablasts or plasma cells in the circulation of patients with highly active and extensive UC at diagnosis. Gut-specific mucosal homing is mainly dependent on the expression of Itg α4β7, a fact that is also therapeutically used in IBD in the form of vedolizumab, an Itg β7 blocking antibody ([45]) Interestingly, post hoc analyses from the GEMINI studies I–III revealed significant lower response and remission rates in anti-TNFα-refractory patients ([46], [47], [48]), a fact that has also been demonstrated for pediatric IBD patients ([49]). In that light, our data suggest that gut-homing a4b7hi plasmablasts are an important feature for successful IFX therapy and vedolizumab treatment with patients displaying low fractions of a4b7hi plasmablasts being more at risk for multiple biological treatment failure. Moreover, intestinal homing toward the terminal ileum is also regulated by the CCR9/CCL25 axis. Here, we observed a strong deficit of CCR9-bearing plasmablasts in both RS and NRS cohorts compared with HC. With regard to our data, the absence of CD39+ plasmablasts in the mucosa may be explained by a decrease in Itg α4β7 and CCR9 which we have observed especially for plasmablasts of NRS patients.

In the analyzed mucosal biopsies, we did not detect relevant numbers of double-stained CD19+ CD73+ B cells, neither in HC nor in IBD patients. Whereas epithelial cells and T cells have been shown to express and up-regulate CD73 in IBD ([41], [50]), CD39 expression is mainly limited to B cells, which led us to the conclusion that the CD39+ B cells in the intestinal mucosa must be plasmablasts or at least memory B cells. Our data show a relative absence of CD39+

---

**Figure 4. ATP degradation capacity and adenosine production by B cells.**
**(A, B)** Overall ATP decrease (A) and the corresponding area under the curve (B) in a luminometric ATP degradation assay with purified B cells of responders (RS; green circles) or nonresponders (NRS; red squares) in comparison with healthy volunteers (HC; white triangles) before initiation of IFX. **(B)** ATP degradation capacity is increased in IFX-responding patients. **(C)** Representative diagram showing the kinetics of ATP (circles) and its products adenosine (squares) and inosine (diamonds) under unstimulated and stimulated conditions. **(D, E, F, G)** Concentrations of adenosine (Ado, (D, F)) and inosine (Ino, (E, G)) at specific time points (150 and 240 min after addition of ATP) under unstimulated (D, E) and stimulated (E, G) conditions. Level of significance is indicated by asterisks: *$P < 0.05$; **$P < 0.01$; ***$P < 0.001$; ****$P < 0.0001$.

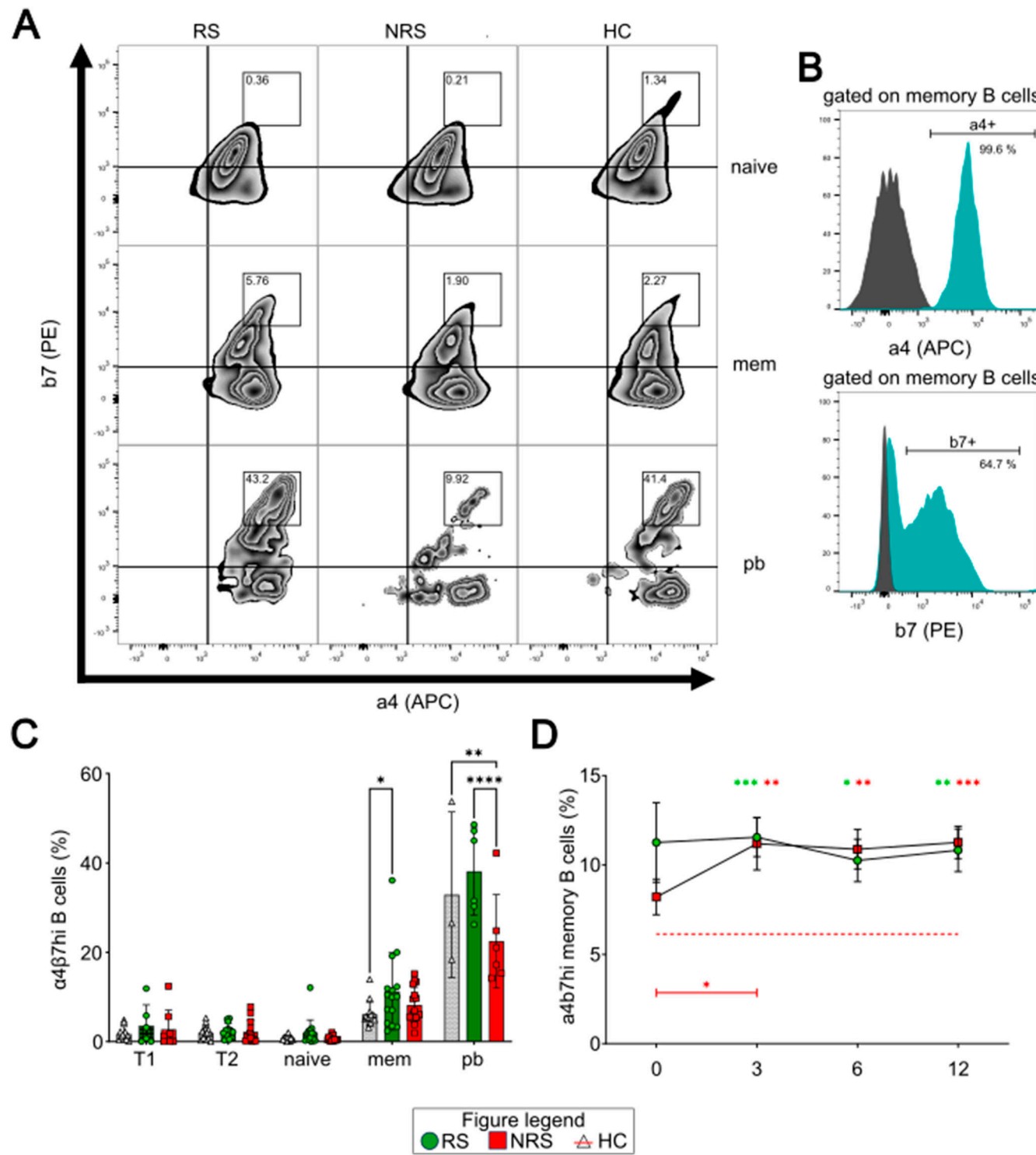

Figure 5. Increased expression of Integrin a4b7 in memory B cells and plasmablasts from IFX-responding IBD patients.
(A) Representative flow cytometric analyses (zebra plots) of the integrin $\alpha4\beta7$ (Itg $\alpha4\beta7$) co-expression on naïve and memory B cells, as well as plasmablasts for each cohort. Squared areas mark the $\alpha4\beta7^{hi}$-expressing subpopulations. (B) Histograms (gated on memory B cells) of isotype control (gray) and stained plots (turquoise) for b7 (upper graph) and a4 (lower graph). (C) Fractions of the Itg $\alpha4\beta7^{hi}$ subsets within all analyzed B-cell subsets before IFX. RS (n = XX): left bars, vertically hatched, circles. NRS (n = XX): middle bars, diagonally hatched, squares. HC: right bars, blank or red dotted line. (D) Fractions of Itg $\alpha4\beta7^{hi}$ memory B cells during IFX therapy. T1: CD24$^{hi}$CD38$^{hi}$ transitional cells; T2: CD24$^{+}$CD38$^{+}$ transitional cells; naïve: CD27$^{-}$ CD38$^{+}$ B cells; mem: CD27$^{+}$ CD38$^{+}$ B cells; pb: CD19$^{+}$ CD20$^{-}$ CD38$^{hi}$ plasmablasts. Level of significance is indicated by asterisks: *$P < 0.05$; **$P < 0.01$; ***$P < 0.001$; ****$P < 0.0001$. The colored asterisks on the top of the plot describe the difference between the specific cohort compared with HC.

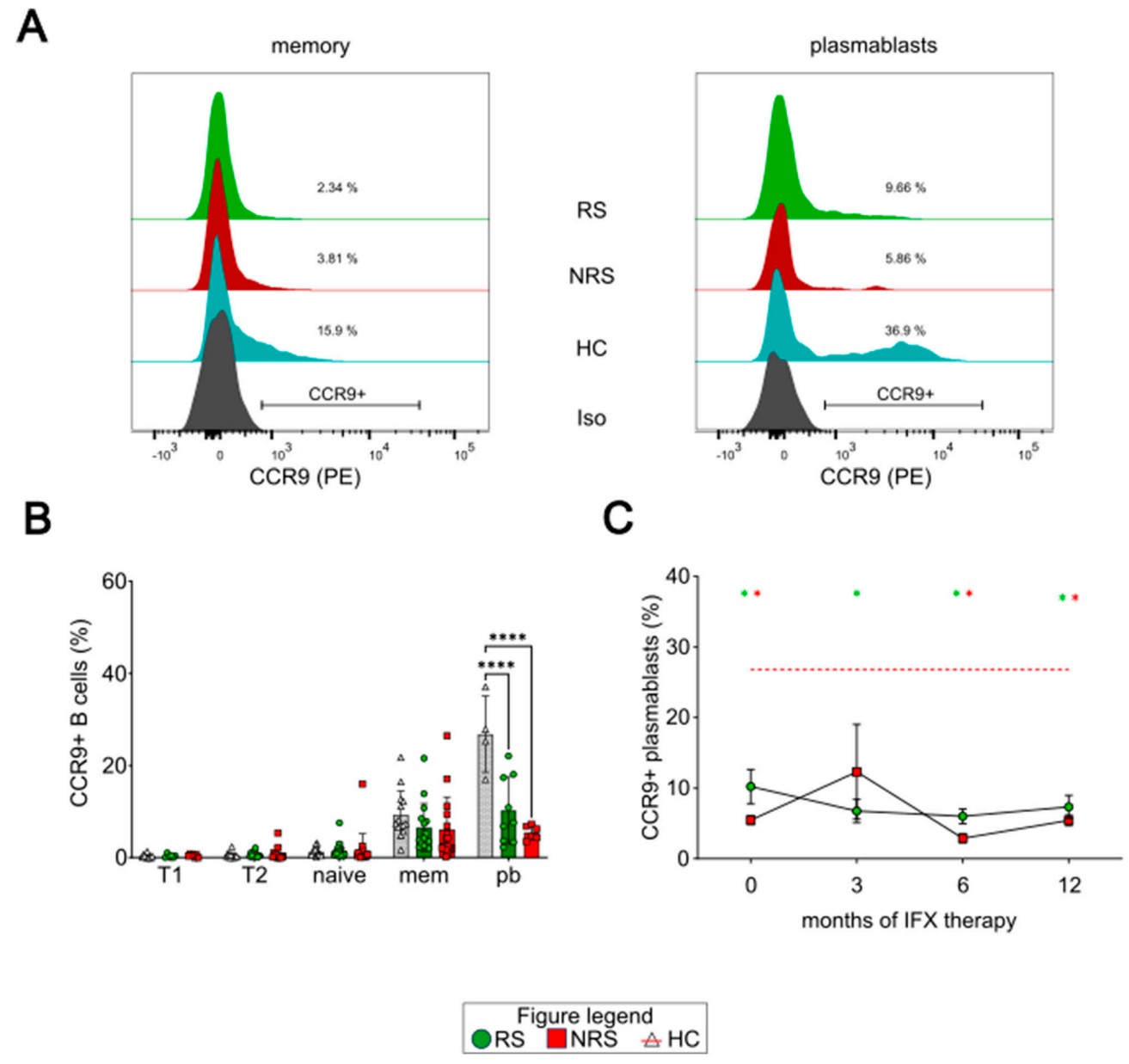

**Figure 6. Expression profile of CCR9 in B cells is altered in pediatric IBD.**
(A) Representative histograms of CCR9 surface abundance on memory B cells (left graph) and plasmablasts (right graph) for each cohort (isotype in gray). (B) Fractions of CCR9$^+$ within all B-cell subsets before IFX. (C) Fractions of CCR9$^+$ plasmablasts during IFX therapy. T1: CD24$^{hi}$CD38$^{hi}$ transitional cells; T2: CD24$^+$CD38$^+$ transitional cells; naïve: CD27$^-$ CD38$^+$ B cells; mem: CD27$^+$ CD38$^+$ B cells; pb: CD19$^+$ CD20$^-$ CD38$^{hi}$ plasmablasts. RS: left bars, vertically hatched, circles. NRS: middle bars, diagonally hatched, squares. HC: right bars, blank or red dotted line. Level of significance is indicated by asterisks: *$P < 0.05$; **$P < 0.01$; ***$P < 0.001$; ****$P < 0.0001$.

plasmablasts in inflamed mucosal biopsies of IBD patients and especially NRS patients before therapy, which consequently might impair the integrity of the functional axis of mucosal ectonucleotidases and the generation of Ado by ATP cleavage (51). From our data, one could also conclude that the distribution of CD39$^+$ mucosal B cells is localization-specific to some extent, with higher the cell numbers, the more distally the biopsy was taken, a fact that might be explained by the differential expression of Itg α4β7 and CCR9. Although in total 56 included subjects and patients represent a considerable sample size for pediatric research, a limitation may be the monocentric approach. Moreover, we would also interpret

the predominance of male patients in the NRS cohort rather as a random than a true finding, as there are no comparable results present in the literature and our study cohort does lack the power to support that finding.

In conclusion, our study provides new valuable insights into the expression and function of immunoregulatory molecules—especially ectonucleotidases in B cells in the context of pediatric IBD. Although IL10-producing plasmablasts were decreased in IFX nonresponders, we detected an up-regulation of CD39 on plasmablasts and increased fractions of CD39/CD73-co-expressing naïve and memory B cells in responding patients. In addition,

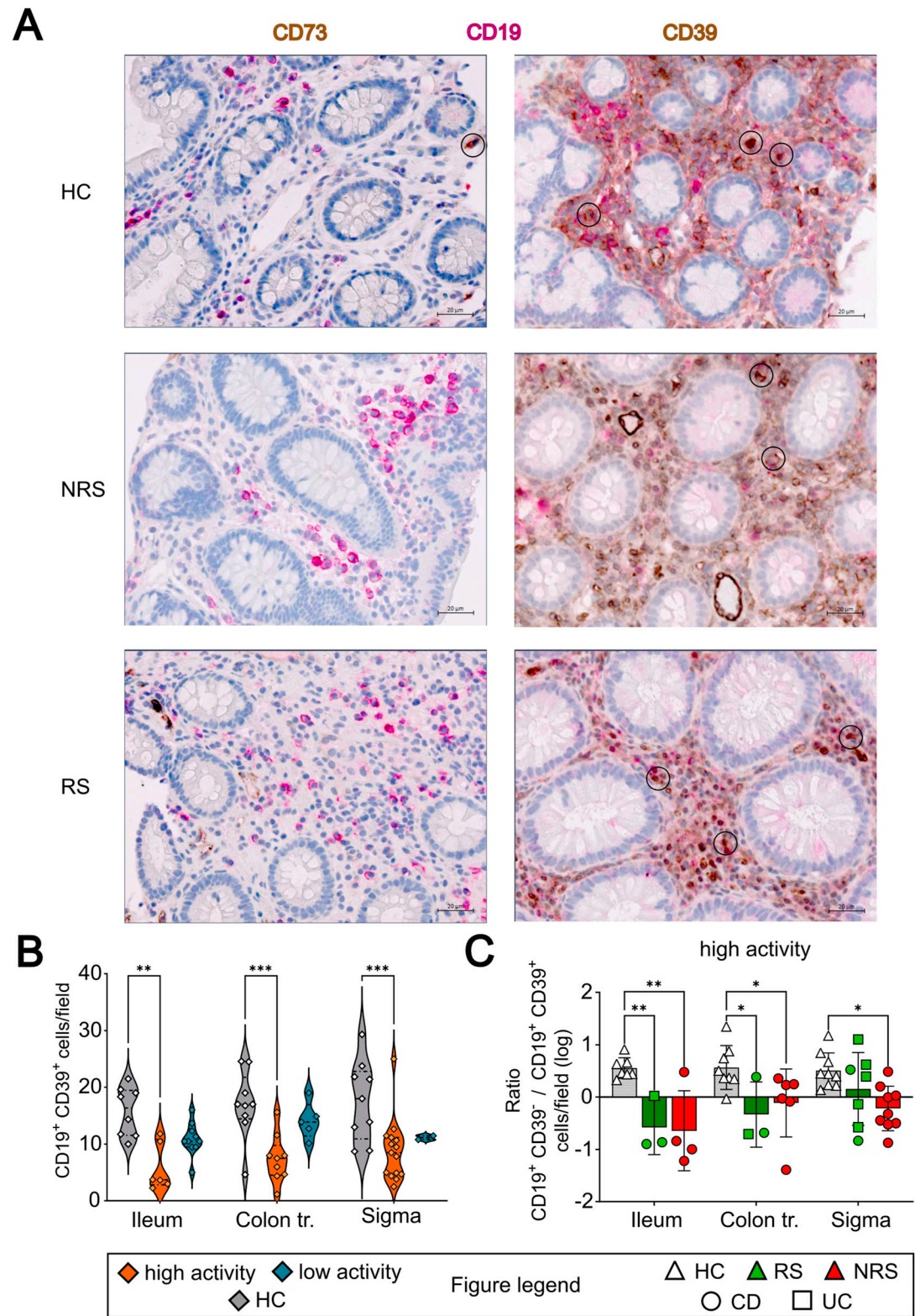

**Figure 7. Abundance of CD39⁺ B cells in the intestinal mucosa is decreased in pediatric IBD.**
**(A)** Representative immunohistochemical stainings of CD19 (pink), CD73 (brown, left side), and CD39 (brown, right side) in mucosal biopsies of healthy volunteers (HC) and future responders (RS) or nonresponders (NRS) before IFX therapy. **(B)** Amount of CD39⁺ B cells in mucosal biopsies with active inflammation (orange) or no inflammation (blue). **(C)** Ratio between CD39⁺ B-cell and overall CD19⁺ B-cell counts in inflamed mucosal biopsies. Level of significance is indicated by asterisks: *$P < 0.05$; **$P < 0.01$; ***$P < 0.001$; ****$P < 0.0001$.

B cells of responders proved to have superior ATP degradation capacities and Ado production compared with NRS and HC. Moreover, IFX nonresponders had a marked deficiency of Itg $\alpha 4\beta 7$hi plasmablasts, whereas both cohorts had fewer CCR9-expressing plasmablasts. Consequently, CD39+ plasmablasts were decreased in biopsies of highly inflamed mucosal tissues, especially in IFX–non-responding IBD patients. Thus, these findings highlight the regulatory potential of CD39- and CD73-expressing B cells in pediatric IBD.

# Materials and Methods

## Patients

For this study, children with IBD after an IFX regimen and age-matched healthy volunteers (HC) were enrolled (Table 1). All patients fulfilled the criteria for Crohn's disease (CD) or ulcerative colitis (UC) with acute exacerbation of disease activity confirmed by endoscopy (EGD and colonoscopy), an increased erythrocyte sedimentation rate, and elevated levels of fecal calprotectin. Inclusion criteria were an age between 2 and 18 yr, a confirmed diagnosis of UC or CD initiation of an IFX therapy, and signed informed consent of the legal custodians. Exclusion criteria were an indeterminate IBD, prednisone therapy >0.1 mg/kg body weight two weeks before enrollment, an already established therapy with IFX or another TNF$\alpha$ inhibitor, past abdominal trauma, participation in other clinical trials for the past 30 d, a malignant disease, concomitant or previous (radio)chemotherapy, infectious enteritis, or pregnancy.

To assess biochemical remission as a result of IFX therapy, we categorized every patient with a fecal calprotectin level below 100 $\mu$g/g after 12 mo as a responder (RS). Patients were also classified as RS if the fecal calprotectin level had decreased to levels <10% of the baseline value at IFX initiation and <250 $\mu$g/g. These cutoffs were selected in line with the ESPGHAN guidelines from 2020, where fecal calprotectin levels are considered a good indicator of therapy response with values below 50 $\mu$g/g representing endoscopic healing, to obtain the optimal threshold for biochemical response (52). In contrast, patients that did not match the responder criteria were labeled as nonresponders (NRS). Another criterion defining nonresponse was the occurrence of colectomy or an in-class change to another TNF$\alpha$-antagonistic regimen (e.g., adalimumab in cases of antidrug antibody production).

## Sample collection

Samples of 9 ml heparinized blood were obtained during each hospital visit for drug infusion (before initiation of the therapy at baseline, after 2–3 mo, at the end of the induction phase, and after 6 and 12 mo during therapy maintenance). PBMCs were immediately isolated via density gradient centrifugation (RotiSep; Carl Roth) and stored in RPMI 1640, 10% FCS, and DMSO in liquid nitrogen for further analysis. Levels of fecal calprotectin were assessed by immunoturbidimetric testing in our hospital's laboratory.

## Flow cytometry

After reconstitution of the frozen samples, thawed PBMCs were immunomagnetically isolated using CD19-specific MACS beads on MS columns (Miltenyi Biotec, Milten).

FACS multicolor analysis was performed using enriched CD19+ cells. Table S1 provides a list of all used antibodies. Before analysis, all antibodies were incubated for 15 min at RT in the dark and washed with phosphate-buffered saline containing 1% FCS. Flow cytometric acquisition was performed using the FACSCanto II apparatus (Becton & Dickinson) and analysis using FlowJo (version 10; FlowJo, LLC). The gating strategy was performed as previously described and shown in Fig 1 (19, 20, 21, 22).

## Cytokine stimulation assay

For the evaluation of cytokine production, immunomagnetic isolated CD19+ cells were cultured per 0.1 × 10^6 in 200 $\mu$l on a 96-well plate for 72 h in RPMI 1640 containing 10% FCS (Gibco), 0.5 $\mu$g/ml CD40 ligand (Miltenyi Biotec), 2 $\mu$M/ml CpG (ODN2006; Miltenyi Biotec), 50 ng/ml rIL21 (Miltenyi Biotec), and 100 U/ml rIL2 (Miltenyi Biotec). After 68 h, cells were washed in PBS containing 1% FCS and restimulated with phorbol 12-myristate 13-acetate and ionomycin, as well as brefeldin A and monensin (BioLegend) for Golgi apparatus blockade. After another 6 h, cells were harvested, washed, and incubated for 15 min at RT in the dark with a Brilliant Blue 515–labeled antibody against CD24, PE-Cy7 anti-CD38, and BV510 anti-CD27, as well as Fixable Viability Stain 780 (Becton & Dickinson) for the exclusion of nonviable cells. After washing, cells were fixed and permeabilized using Fix&Perm Cell Fixation and Permeabilization Kit (Thermo Fisher Scientific). Intracellular cytokines were stained with a PE-labeled antibody against IL10, PerCP-Cy5.5–labeled antibody against TNF$\alpha$, and an APC-labeled antibody against GrzmB. Intracellular antibodies were incubated for 20 min at RT in the dark, washed, and instantly analyzed by flow cytometry.

## Histology

Immunohistochemical staining was performed on formalin-fixed and paraffin-embedded tissue samples, sectioned into 1-$\mu$m slices. Primary antibodies used were CD19 (mouse, anti-human; OriGene), and CD39 and CD73 (rabbit, anti-human; Abcam). Biotinylated antibodies were used for secondary staining (Vector). Staining was performed with a 1:200 dilution of streptavidin-HRP or streptavidin-AP for 30 min at RT, and the slides were counterstained with hematoxylin. For color development, the slides were incubated with 3,3'-diaminobenzidine (DAB) and Fast Red.

## Assessment of ATP degradation capacity of B cells

For assessment of ATP degradation, CD19+ purified B cells from IBD patients after 12 mo of therapy and HC were used. B cells were separated using magnetic immunobeads and stimulated as described previously. For detection of ATP hydrolysis, 10,000 purified B cells were incubated in 200 $\mu$l of cell culture buffer (10 mM glucose, 20 mM Hepes, 5 mM KCl, 12 mM NaCl, 2 mM CaCl$_2$, 5 mM tetramisole) in 96-well LoBind plates in the presence of 20 $\mu$M ATP

for various time periods. Control wells contained cells or ATP alone. All experiments were performed in duplicates. Analysis of ATP degradation because of enzymatic activity of CD73 and CD39 was performed using a luciferase-based assay. At time points 5, 30, 45, and 60 min, luciferin/luciferase was added to Veritas Microplate Luminometer and relative light units were detected. As a standard curve, a given quantity of ATP was diluted from 1 $\mu$M down to 0.1 pM to allow the calculation of ATP concentrations of the samples. Cells were further incubated for 60, 90, 120, 150, and 240 min; at those time points, supernatants are collected and stored at –80°C for mass spectrometry. For stimulated conditions, the same stimulation agents were used as described above. Percent ATP degradation was normalized by dividing with a factor/quotient based on the means of all blank ATP measurements and the blank ATP means of the specific experiment day.

### Mass spectrometry

Inosine and adenosine were measured using high-pressure liquid chromatography–tandem mass spectrometry (HPLC-MS/MS) by selected reaction monitoring with guanosine as the internal standard. Samples were injected into an Acquity ultra-performance liquid chromatographic system (Waters) and were separated on a C18 column (Waters UPLC BEH C18; 1.7 $\mu$m; 2.1 × 100 mm) using the following elution conditions: mobile phase A, 1% acetic acid in $H_2O$; mobile phase B, methanol; flow rate, 0.3 ml/min; elution gradient (A/B) was 99.5%/0.5% (0–2 min), 98%/2% (2–3 min), 85%/15% (3–4 min), and 99.5%/0.5% (4–5 min). Purine levels were analyzed with a TSQ Quantum Ultra triple quadrupole mass spectrometry equipped with a heated electrospray ionization source. The mass spectrometer was operated in the positive-ion mode, and the following mass-to-charge transitions were monitored: 348→136 for 5'-AMP; 268→136 for ADO.

### Statistical analysis

Statistical analysis was performed using GraphPad Prism software (version 10; GraphPad). Data at specific points in time were tested for normal distribution and analyzed using either one-way ANOVA or Kruskal–Wallis tests.

Longitudinal courses were tested for differences between points in time, diseases, or interactions of both using a mixed model with the Geisser–Greenhouse correction as implemented in GraphPad Prism 8.0. For multiple comparisons, either Tukey's (baseline) or Dunnett's multiple comparisons test was used. For indication of significance levels, asterisks are used: *, $P < 0.05$; **, $P < 0.01$; ***, $P < 0.001$; ****, $P < 0.0001$.

In figures showing specific points in time, the SD is plotted, whereas for graphs depicting longitudinal data, only SEM is displayed for clarity reasons.

### Study approval

The study was approved by the Scientific Ethics Committee of the Friedrich-Alexander University Erlangen-Nuremberg (#347_15B); all participants provided informed consent before study enrollment.

## Data Availability

The original data underlying this article will be shared on reasonable request to the corresponding author.

## Supplementary Information

## Acknowledgements

The authors would like to thank Perdita Weller and Tatjana Flamann, for technical assistance. We would like to thank the patients and their parents for participating in this study. This work was supported by the *Frieda-Marohn-Stiftung* (Grant number: PMTNFA) granted to A Hoerning and a research grant from Hexal AG, Holzkirchen, granted to A Hoerning and A Schnell. The Interdisciplinary Center for Clinical Research (IZKF) of the Friedrich-Alexander-Universität (FAU) and the University Hospital funded this project (Laboratory Rotation and Clinician Scientist Program (CSP) to A Schnell).

### Author Contributions

A Schnell: conceptualization, data curation, formal analysis, investigation, visualization, methodology, project administration, and writing—original draft, review, and editing.
B Schwarz: conceptualization, data curation, formal analysis, investigation, and methodology.
H Schmidt: investigation.
I Allabauer: data curation, validation, investigation, and methodology.
W Schuh: conceptualization, formal analysis, supervision, and writing—review and editing.
AP Regensburger: conceptualization, supervision, visualization, and writing—review and editing.
M Rauh: investigation and methodology.
J Woelfle: conceptualization and writing—review and editing.
A Hoerning: conceptualization, supervision, funding acquisition, project administration, and writing—original draft, review, and editing.

### Conflict of Interest Statement

A Schnell has received travel expenses from Ipsen Pharma and Mirum Pharmaceuticals. A Hoerning has received research grants for clinical studies, speaker's fees, honoraria, or travel expenses from AbbVie, Ipsen Pharma, Mirum Pharmaceuticals, MSD, Novartis, Danone, Sanofi, Pfizer, and Takeda. All other authors have no competing interests to declare.

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
