## [Reviewer comments · Life Science Alliance]

Life Science Alliance

Adenosine-generating CD39+ plasmablasts predispose to successful Infliximab therapy in pediatric IBD

Alexander Schnell, Benedikt Schwarz, Hannah Schmidt, Ida Allabauer, Wolfgang Schuh, Adrian Regensburger, Manfred Rauh, Joachim Wölfle, and André Hörning

DOI: <https://doi.org/10.26508/lsa.202403055>

Corresponding author(s): Alexander Schnell, Universitätsklinikum Erlangen

Review Timeline:

Submission Date:	2024-09-20
Editorial Decision:	2024-12-05
Revision Received:	2025-02-27
Editorial Decision:	2025-02-28
Revision Received:	2025-03-18
Accepted:	2025-03-18

Transaction Report:

December 5, 2024

Re: Life Science Alliance manuscript #LSA-2024-03055-T

Dr. Alexander Schnell
Universitätsklinikum Erlangen
Department of Paediatrics and Adolescent Medicine
Loschgstraße 15
Erlangen 91054
Germany

Dear Dr. Schnell,

Thank you for submitting your manuscript entitled "Adenosine-generating CD39+ plasmablasts predispose to successful Infiximab therapy in pediatric IBD" to Life Science Alliance. The manuscript was assessed by expert reviewers, whose comments are appended to this letter. We invite you to submit a revised manuscript addressing the Reviewer comments.

Thank you for this interesting contribution to Life Science Alliance. We are looking forward to receiving your revised manuscript.

Sincerely,

B. MANUSCRIPT ORGANIZATION AND FORMATTING:

Reviewer #1 (Comments to the Authors (Required)):

Schnell et al. explore the role of CD39+ plasmablasts and other B-cell subsets in predicting and enhancing the efficacy of Infliximab (IFX) therapy in children with Inflammatory Bowel Disease (IBD).

Key findings include:

1. Higher Expression of CD39 in Responders: Children who responded well to IFX treatment had higher levels of CD39+ plasmablasts and CD39/CD73 co-expressing B-cells, which play a role in breaking down ATP to produce adenosine, a molecule with immunosuppressive effects.
 2. Pro-inflammatory B-cell Profile in Non-responders: Those who did not respond showed fewer IL10-producing (anti-inflammatory) B-cells and a predominance of TNF α -producing (pro-inflammatory) B-cells, indicating an inflammatory profile that IFX therapy could not fully suppress.
 3. Gut-specific Homing Receptors: A higher number of IFX-responsive patients had B-cells expressing gut-homing markers (Itg $\alpha 4\beta 7$ and CCR9), which may allow these cells to reach and regulate inflamed gut tissue more effectively.
 4. Mucosal B-cell Characteristics: Non-responders also had fewer CD39+ B-cells in the inflamed gut tissue, suggesting these cells' deficiency may impact local immune regulation negatively.
- The study suggests that CD39+ plasmablasts and ATP-degrading B-cells are potentially critical in facilitating a positive IFX response, offering insights into targeted therapies for pediatric IBD.

Major Comments

1. B cell population comparisons: The authors report differences in B cell population composition between patients and healthy donors. Were the healthy donors age-matched? Please provide demographic details for the healthy donor group in a table, as age differences could account for some of the observed variation.
2. Supplementary Figure 1 - Identification markers: CD27 should be used to identify memory B cells in Supplementary Figure 1. Additionally, why are CD38-negative, CD24-positive cells not considered in this figure?
3. Gating strategy for memory B cells and plasmablasts: Could the authors display CD24 versus CD27 to clarify the gating of memory B cells, plasmablasts, and plasma cells?
4. Figure order and text alignment: The figures appear out of sequence; for instance, Figures 5A and 5B are mentioned before Figure 4 in the text. Additionally, the results described in the text (line 296) for Figures 5A and 5B do not correspond with the figures themselves.
5. Figure 2B - Discrepancy in dot counts: In Figure 2B, the number of dots for plasmablasts is inconsistent, with only 8 dots instead of 9. Could the authors clarify this discrepancy?
6. T1 and T2 populations: The authors should define T1 and T2 populations, and the gating strategy for these populations should include CD21 as a marker.
7. Figure 3C - Memory population visibility: In Figure 3C, the double-positive memory population is unclear in the zebra plot and does not align with the bar plot results. Improved clarity is needed for this figure.
8. Figure 5B - Gating details: On what population is Figure 5B's histogram gated? Please specify if it is gated on B cells, memory cells, or plasmablasts.
9. Figure 5D - Asterisk explanation: In Figure 5D, does the asterisk indicate a significant difference from the healthy donor (HD) group? This needs clearer explanation.
10. Supplementary Figure 2 - Inconsistent dot counts: In Supplementary Figure 2A, the Ileum high-activity group has only 4 dots, whereas in panel B, it has 5 dots. Please address this inconsistency.
11. Results interpretation (line 301-303): How do the authors explain that responders are less similar to healthy controls (HC) than non-responders (NRS)?
12. Results interpretation (line 343-345): Why are HC more similar to NRS than responders in this context? Further explanation is required.
13. Data omissions: Numerous data points are described but not shown. Please include all relevant data in the manuscript.
14. Definition of high and low activity: The terms "high activity" and "low activity" are not defined. A clear definition is necessary for reader comprehension.
15. CD39 labeling: Showing CD39-negative data is redundant since it is the reciprocal of CD39-positive data. Consider omitting it.
16. Therapeutic implications and IFX response: The authors suggest that for effective therapy with infliximab (IFX), plasmablasts

(CD39-high) should home to the gut. However, anti-beta-7 integrin therapy, which blocks plasmablast gut homing, is a known therapeutic approach for inflammatory bowel disease. This seems contradictory and should be reconciled with the current findings.

Minor Comments

1. Abbreviation clarity: The abstract contains numerous unexplained abbreviations. Please define all abbreviations upon first use.
2. Abstract clarity (line 37): The abstract suggests that healthy donors (HD) undergo IFX treatment, which is presumably incorrect. Please revise for clarity.
3. Participant enrollment: In line 95, the sentence would benefit from the addition of "were enrolled" at the end.
4. Gating strategy accuracy: The CD24/CD38 gating strategy appears unusual. Are the axis labels correct?
5. Citation alignment: None of the cited papers provide the exact B cell gating strategy used. Please ensure that cited references accurately reflect the methods employed.
6. Figure 6A format: Figure 6A is described as a FACS plot but is actually a histogram. Please correct this discrepancy.

Reviewer #2 (Comments to the Authors (Required)):

1) B cells play a key immunoregulatory role, including producing interleukin-10 (IL-10) and expressing ectonucleotidases like CD39 and CD73, which help maintain immune balance by breaking down eATP into immunosuppressive adenosine. This study examined the expression of these molecules on B cells in pediatric IBD patients and their association with infliximab (IFX) treatment outcomes.

IFX responders showed increased CD39 expression on plasmablasts and higher levels of CD39/CD73 co-expression on naïve and memory B cells, along with enhanced ATP degradation and adenosine production before treatment. Response was associated with increased IL-10-producing plasmablasts and $\alpha 4\beta 7$ hi plasmablasts.

These findings provide for the first time evidence of the importance of CD39/CD73 in B cells in pediatric IBD and suggest that CD39 might be a biomarker for IFX response.

2) Data are strongly supportive of all the main points of the papers

3) because of the known relationship between IgA/IgG plasma blast in IBD (PMID: 36752797; PMID: 35190725) and the known role of IL-10 in IgA+ plasmablast/plasma cells (PMID: 30612739), I feel the paper will increase in impact if the authors could at least try to subste the expression of CD49/CD73 in RS/NRS according to the B cell isotype.

Manuscript ID: LSA-2024-03055-T

Title: "Adenosine-generating CD39+ plasmablasts predispose to successful Infliximab therapy in pediatric IBD"

Dear LSA editorial team,

On behalf of all authors I would like to thank you and the referees for the review and their interest of our manuscript mentioned above.

Please find below our point-to-point response to the reviewers' criticisms and suggestions. For your convenience, we first repeat the reviewer's comment (in italics) followed by our reply.

Reviewer #1 (Comments to the Authors (Required)):

Major Comments

1. B cell population comparisons: The authors report differences in B cell population composition between patients and healthy donors. Were the healthy donors age-matched? Please provide demographic details for the healthy donor group in a table, as age differences could account for some of the observed variation.

Reply: Indeed, the healthy probands were age matched, we have provided their demographic data in Table 1.

2. Supplementary Figure 1 - Identification markers: CD27 should be used to identify memory B cells in Supplementary Figure 1. Additionally, why are CD38-negative, CD24-positive cells not considered in this figure?

Reply: In preliminary experiments, CD27 showed a poor staining behaviour on stimulated B cells and did not help to discriminate memory B cells from other subsets. We omitted the CD38-negative population as it was not clear to us from which cells this population consists of.

3. Gating strategy for memory B cells and plasmablasts: Could the authors display CD24 versus CD27 to clarify the gating of memory B cells, plasmablasts, and plasma cells?

Reply: As you can see in the attached figure, the Bmem subset can be adequately determined based on CD27, with CD24+ CD27+ cells being just a further subset of B cells.

4. Figure order and text alignment: The figures appear out of sequence; for instance, Figures 5A and 5B are mentioned before Figure 4 in the text. Additionally, the results described in the text (line 296) for Figures 5A and 5B do not correspond with the figures themselves.

Reply: We have corrected the order and alignment of the respective figures.

5. Figure 2B - Discrepancy in dot counts: In Figure 2B, the number of dots for plasmablasts is inconsistent, with only 8 dots instead of 9. Could the authors clarify this discrepancy?

Due to the small SD for these data points and the scale of the plot, all data points appear on a very narrow space. If you zoom in, you can count nine data points.

6. T1 and T2 populations: The authors should define T1 and T2 populations, and the gating strategy for these populations should include CD21 as a marker.

Reply: We agree with the reviewer that additional markers would have helped to better discriminate T1 and T2. However, due to the limited amount of channels for FACS analysis, we felt that CD24/CD38 would provide sufficient information for T1/T2 discrimination. Especially with zebra blot depiction, in every patient two distinct populations along the characteristic CD24/CD38 „nose“ was delimitable.

7. Figure 3C - Memory population visibility: In Figure 3C, the double-positive memory population is unclear in the zebra plot and does not align with the bar plot results. Improved clarity is needed for this figure.

Reply: We would like the reviewer for bringing up this point. Indeed, there was a malrotation of some plots. We have created a new version which now clarifies all important findings.

8. Figure 5B - Gating details: On what population is Figure 5B's histogram gated? Please specify if it is gated on B cells, memory cells, or plasmablasts.

Reply: This was gated on memory cells, we have indicated this in the manuscript.

9. Figure 5D - Asterisk explanation: In Figure 5D, does the asterisk indicate a significant difference from the healthy donor (HD) group? This needs clearer explanation.

Reply: Indeed, the coloured asterixes on the top of the plot describe the difference between the specific cohort compared to HC. We have clarified this in the Figure legend.

10. Supplementary Figure 2 - Inconsistent dot counts: In Supplementary Figure 2A, the Ileum high-activity group has only 4 dots, whereas in panel B, it has 5 dots. Please address this inconsistency.

Reply: We would like to thank the reviewer for bringing up this inconsistency. We checked the data sheets again and indeed there was a data insertion error for Supp Fig 2. We have corrected this error.

11. Results interpretation (line 301-303): How do the authors explain that responders are less similar to healthy controls (HC) than non-responders (NRS)?

Reply: We believe the up-regulation of CD39/CD73 in IFX-responders to be a compensatory mechanism that sustains remission. In NRS, this mechanism appears to happen less pronouncedly. HC lack systemic inflammation and have therefore no need for a compensatory up-regulation of ectonucleotidases.

12. Results interpretation (line 343-345): Why are HC more similar to NRS than responders in this context? Further explanation is required.

Reply: See for point 11.

13. Data omissions: Numerous data points are described but not shown. Please include all relevant data in the manuscript.

Reply: Data points were omitted when the results were deemed worth of reporting but were not relevant with regard to the major findings of our study. Thus, we decided to show only the most relevant findings for clarity reasons.

14. Definition of high and low activity: The terms "high activity" and "low activity" are not defined. A clear definition is necessary for reader comprehension.

Reply: All specimens were marked as high activity where routine pathological work-up revealed signs of active inflammation (presence of neutrophils) or chronicity (presence of lymphoplasmacellular infiltrations).

15. CD39 labeling: Showing CD39-negative data is redundant since it is the reciprocal of CD39-positive data. Consider omitting it.

Reply: We gather that the reviewer is referring to Figure 7C. In our opinion, this finding is an important feature that clearly distinguishes IBD patients from HC as there seems to be a dysbalance of CD19⁺ CD39⁺ vs CD19⁺ CD39⁻ B cells in inflamed mucosa.

16. Therapeutic implications and IFX response: The authors suggest that for effective therapy with infliximab (IFX), plasmablasts (CD39-high) should home to the gut. However, anti-beta-7 integrin therapy, which blocks plasmablast gut homing, is a known therapeutic approach for inflammatory bowel disease. This seems contradictory and should be reconciled with the current findings.

Reply: We have inserted the following passage into our manuscript: Gut-specific mucosal homing is mainly dependent on the expression of Itg $\alpha 4\beta 7$, a fact that is also therapeutically used in IBD in the form of vedolizumab, an Itg $\beta 7$ blocking antibody [45] Interestingly, post-hoc analyses from the GEMINI studies I – III revealed significant lower response and remission rates in anti-TNF α -refractory patients, a fact that has also been demonstrated for pediatric IBD patients. In that light, our data suggest that gut-homing $\alpha 4\beta 7$ hi plasmablasts are an important feature for successful IFX therapy as well as Vedolizumab treatment with patients displaying low fractions of $\alpha 4\beta 7$ hi plasmablasts being more at risk for multiple biological treatment failure.

Minor Comments

1. Abbreviation clarity: The abstract contains numerous unexplained abbreviations. Please define all abbreviations upon first use.

Reply: Definitions for all relevant abbreviations are now included.

2. Abstract clarity (line 37): The abstract suggests that healthy donors (HD) undergo IFX treatment, which is presumably incorrect. Please revise for clarity.

Reply: We have revised this point.

3. Participant enrollment: In line 95, the sentence would benefit from the addition of "were enrolled" at the end.

Reply: We have revised this point.

4. Gating strategy accuracy: The CD24/CD38 gating strategy appears unusual. Are the axis labels correct?

Reply: The axis labels are correct, we have used this gating strategy. Furthermore, we included references for this specific gating strategy.

5. Citation alignment: None of the cited papers provide the exact B cell gating strategy used. Please ensure that cited references accurately reflect the methods employed.

Reply: We have included several references that reflect the used gating strategy.

6. Figure 6A format: Figure 6A is described as a FACS plot but is actually a histogram. Please correct this discrepancy.

Reply: We have revised this point.

Reviewer #2 (Comments to the Authors (Required)):

1) B cells play a key immunoregulatory role, including producing interleukin-10 (IL-10) and expressing ectonucleotidases like CD39 and CD73, which help maintain immune balance by breaking down eATP into immunosuppressive adenosine. This study examined the expression of these molecules on B cells in pediatric IBD patients and their association with infliximab (IFX) treatment outcomes.

IFX responders showed increased CD39 expression on plasmablasts and higher levels of CD39/CD73 co-expression on naïve and memory B cells, along with enhanced ATP degradation and adenosine production before treatment. Response was associated with increased IL-10-producing plasmablasts and $\alpha 4\beta 7$ hi plasmablasts.

These findings provide for the first time evidence of the importance of CD39/CD73 in B cells in pediatric IBD and suggest that CD39 might be a biomarker for IFX response.

2) Data are strongly supportive of all the main points of the papers

3) because of the known relationship between IgA/IgG plasma blast in IBD (PMID: 36752797; PMID: 35190725) and the known role of IL-10 in IgA+ plasmablast/plasma cells (PMID: 30612739), I feel the paper will increase in impact if the authors could at least try to subset the expression of CD49/CD73 in RS/NRS according to the B cell isotype.

Reply: We thank the reviewer for this important commentary. We are planning to perform the suggested experiments in the future. At the moment however, all samples from the included patients have been used for the shown experiments.

February 28, 2025

RE: Life Science Alliance Manuscript #LSA-2024-03055-TR

Dr. Alexander Schnell
Universitätsklinikum Erlangen
Department of Paediatrics and Adolescent Medicine
Loschgestraße 15
Erlangen 91054
Germany

Dear Dr. Schnell,

Thank you for submitting your revised manuscript entitled "Adenosine-generating CD39+ plasmablasts predispose to successful Infliximab therapy in pediatric IBD". We would be happy to publish your paper in Life Science Alliance pending final revisions necessary to meet our formatting guidelines.

- please be sure that the authorship listing and order is correct
- please upload your main and supplementary figures as single files
- please add the X and Bluesky handles of your host institute/organization as well as your own or/and one of the authors in our system
- please note that titles in the system and manuscript file must match
- please be sure that the authorship listing and order are correct and match between the system and manuscript file
- please consult our manuscript preparation guidelines <https://www.life-science-alliance.org/manuscript-prep> and make sure your manuscript sections are in the correct order
- please upload manuscript file without track changes
- please incorporate any points from the Conclusion section into the Discussion; we only allow a Discussion section
- the contributions selected for Wolfgang Schuh, Adrian P. Regensburger, and Joachim Wölfle do not qualify them for authorship. Please either update the contributions in our system and the Author Contributions section of the manuscript or let us know if the authors need to be removed (and added eventually to the acknowledgment section)
- please add your main, supplementary figure, and table legends to the main manuscript text after the references section;
- There is a callout for figure 8C, and there are only 7 figures provided -- please correct
- please add a callout for Figure 1C to your main manuscript text

FIGURE CHECK:

- please add scale bars to Figure 7A

LSA now encourages authors to provide a 30-60 second video where the study is briefly explained. We will use these videos on social media to promote the published paper and the presenting author (for examples, see <https://docs.google.com/document/d/1-UWCfbE4pGcDdcgzcmiuJI2XMBJnxKYeqRvLLrLS08s/edit?usp=sharing>). Corresponding or first-authors are welcome to submit the video. Please submit only one video per manuscript. The video can be emailed to contact@life-science-alliance.org

A. FINAL FILES:

B. MANUSCRIPT ORGANIZATION AND FORMATTING:

Sincerely,

March 18, 2025

RE: Life Science Alliance Manuscript #LSA-2024-03055-TRR

Dr. Alexander Schnell
Universitätsklinikum Erlangen
Department of Paediatrics and Adolescent Medicine
Loschgestraße 15
Erlangen 91054
Germany

Dear Dr. Schnell,

Thank you for submitting your Research Article entitled "Adenosine-generating CD39+ plasmablasts predispose to successful Infliximab therapy in pediatric IBD". It is a pleasure to let you know that your manuscript is now accepted for publication in Life Science Alliance. Congratulations on this interesting work.

DISTRIBUTION OF MATERIALS:

Again, congratulations on a very nice paper. I hope you found the review process to be constructive and are pleased with how the manuscript was handled editorially. We look forward to future exciting submissions from your lab.

Sincerely,
